# Pre-Molten, Wet, and Dry Molten Globules en Route to the Functional State of Proteins

**DOI:** 10.3390/ijms24032424

**Published:** 2023-01-26

**Authors:** Munishwar Nath Gupta, Vladimir N. Uversky

**Affiliations:** 1Department of Biochemical Engineering and Biotechnology, Indian Institute of Technology, Hauz Khas, New Delhi 110016, India; 2Department of Molecular Medicine and USF Health Byrd Alzheimer’s Research Institute, Morsani College of Medicine, University of South Florida, Tampa, FL 33612, USA

**Keywords:** baroenzymology, cryoenzymology, intrinsically disordered proteins, macromolecular crowding, nanomaterials, partially folded intermediate, protein engineering, protein flexibility, protein folding, protein structure

## Abstract

Transitions between the unfolded and native states of the ordered globular proteins are accompanied by the accumulation of several intermediates, such as pre-molten globules, wet molten globules, and dry molten globules. Structurally equivalent conformations can serve as native functional states of intrinsically disordered proteins. This overview captures the characteristics and importance of these molten globules in both structured and intrinsically disordered proteins. It also discusses examples of engineered molten globules. The formation of these intermediates under conditions of macromolecular crowding and their interactions with nanomaterials are also reviewed.

## 1. Introduction

In the classical picture of enzymology, the native structure of a protein is intimately correlated to its function [1], and the functional 3D structure of proteins is determined solely by their amino acid sequences [2,3,4,5]. A deviation from the native structure accompanied by the loss of biological activity was defined as protein denaturation. Hence, study of the process of the unfolding of a protein molecule (under various denaturing conditions) was as responsible for gaining crucial knowledge on the structure-function relationships in proteins as investigating protein refolding. In the three somewhat related phenomena, protein folding (i.e., spontaneous formation of a 3D structure by the nascent polypeptides in the cell), protein unfolding, and protein refolding from the unfolded state in the test tube, the transition between native structure and unfolded/denatured structure(s) was the common thread [6].

The transition between the native and denatured states of small globular proteins was initially considered a two-state process (two-state model of protein unfolding) [7,8,9]. Over the years, two generic folding intermediates were identified: the molten globule (MG) [10,11,12,13,14,15,16,17,18,19,20,21] and the pre-molten globule (PMG) [22,23,24,25,26]. Curiously, the existence of such folding intermediates was predicted in 1973 by Oleg B. Ptitsyn (1929–1999) based on the theoretical considerations of the potential mechanisms by which the hierarchical structure of a native globular protein can be rapidly formed, despite the astronomically large number of possibilities by which a polypeptide chain can be packed into a compact globule [27]. More recently, the concepts of “wet” and “dry” molten globules have emerged, where dry molten globular (DMG) intermediates represent an expanded form of the native protein with a dry core [28,29,30,31,32,33,34,35,36,37,38]. This is an interesting observation, as early studies indicated that the molten globule represents a highly hydrated state, with water inside the molten globule interior possessing characteristics of a highly associated liquid [21]. However, as early as in 1989, theoretical analysis of the denatured states of globular proteins suggested that since the compactness of a denatured protein may vary within a wide range, several denatured forms can be distinguished, such as coil, swollen globule, the “wet” molten globule (the compact state with pores occupied by solvent), and the “dry” molten globule where solvent does not penetrate inside the protein [39].

The current (at least) five-state picture of protein folding can be schematically depicted as following:U ↔ PMG ↔ WMG ↔ DMG ↔ N

This overview captures the evolving nature of our understanding of the protein folding process in terms of this scheme. A brief discussion on these various forms of MGs is included. It also covers the interface of protein folding with macromolecular crowding and intrinsic disorder in proteins [40,41].

## 2. Molten Globule as an (Un)Folding Intermediate

Historically, protein denaturation and unfolding studies are based on the well-accepted and rather obvious (at least now) mantra stating, “Structure does exit since it can be broken”. These studies played crucial roles in establishing protein science in general and in understanding the basis of the correlation between protein amino acid sequence and function in particular. As early as in 1931, Hsien Wu (1893–1959) proposed the first theory of protein denaturation: the active structure is known to exist because it is destroyed by denaturation [42,43]. His paper published in the *Chinese Journal of Physiology* contained the first statement that protein function depends on prior structure [42,43]. However, even earlier, in 1925, Mortimer Louis Anson (1901–1968) and Alfred Ezra Mirsky (1900–1974) showed that intact hemoglobin can exist as such near the neutral point only, whereas dilute acid or alkali changed it to the denatured form, which could fold back to its native state upon restoration of native conditions, indicating that protein denaturation and unfolding are reversible processes [44].

In 1936, the first Western review on protein denaturation that represents the first modern theory of native and denatured proteins was published, where Alfred Mirsky and Linus Pauling (1901–1994) stated that the loss of certain highly specific properties constitutes the most significant change that occurs in the denaturation of a native protein [45]. By 1944, it became clear that native proteins have unique structures, that the denaturation processes are manifold in nature and magnitude, and that the addition of high concentrations of strong denaturants, such as guanidine hydrochloride (GdmHCl) or urea, to a protein causes a complete (or almost complete) disruption of all conformational interactions and, as consequence, to the transformation of a protein molecule into the highly disordered state of a random coil [46]. Furthermore, the authors of this seminal review stated: “The term denaturation has been used rather loosely and indiscriminately to denote ill-defined changes in the properties of proteins, caused by a variety of chemical, physical, and biological agents. The observation that many unrelated processes may cause similar changes in a protein early led to the belief that any single change, such as the formation of a coagulum, suffices to characterize a ‘denatured’ protein, and that all denaturing agents are alike in their action. Although proteins are now known to respond differently to various kinds of denaturation, the supposition of the singleness of the denaturation process has persisted” [46].

They also defined denaturation as “any non-proteolytic modification of the unique structure of a native protein, giving rise to definite changes in chemical, physical, or biological properties” [46]. It is obvious that a clear distinction should be made between the terms “denaturation” and “unfolding”. Here, as defined above, denaturation is a process leading to the elimination of protein functionality resulting from the disruption of functional 3D structure. This can be triggered by a wide range of conditions, with the resulting denatured forms possessing a wide spectrum of properties depending on the conditions in which they were achieved. On the contrary, protein unfolding is defined as a process leading to the complete elimination of all the conformational forces stabilizing the native protein structure, resulting, therefore, in the formation of a coil-like conformation.

Retrospectively, finding partially folded species of globular proteins under a variety of denaturing conditions should not be surprising. This is because the unique 3-D structure of a protein molecule is stabilized by specific non-covalent interactions, such as hydrogen bonds, hydrophobic interactions, electrostatic interactions and salt bridges, and van der Waals interactions. Since these conformational forces have different physical natures, it is quite possible that they would react differently to the changes in the environment, where under specific conditions, some forces would decline and dissipate, whereas others would stay unchanged or even strengthen. In these cases, the protein molecule is obviously losing its biological activity; i.e., it is becoming denatured, but since not all the conformational forces are “shutdown”, denaturation is not necessarily accompanied by the complete unfolding of a protein, giving rise to the appearance of new conformations with properties halfway between those of native and completely unfolded states. Therefore, various degrees of denaturation/unfolding must exist, depending on the extent to which the structure of the protein has been modified under given conditions. Clearly, the fact that the extent of denaturation can be different is incompatible with the “all-or-none” hypothesis that a given protein can exist in only one of two states, the completely native or the completely denatured/unfolded [46].

These important considerations were rooted in the experimental evidence accumulated in the 1930s and 1940s, when the incomplete unfolding and existence of some intermediate stages of denaturation were recognized in several instances [47,48,49,50,51]. Furthermore, as follows from later studies, some denatured forms produced at milder denaturing conditions (e.g., heat- or pH-denatured proteins) can undergo additional structural alterations in the presence of strong denaturants, such as urea or GdmHCl [52]. Therefore, since the final denatured conformations of proteins are strongly dependent on the denaturing agent, not all denatured states are structurally similar, and under certain conditions the protein molecules are not completely unfolded.

These very logical conclusions were formulated in a classical review by Charles Tanford (1921–2009) [9], which was one of the first papers providing in-depth analysis of the possibility that during the unfolding of globular proteins, accumulation of some equilibrium intermediate states might be expected. Unfortunately, since the results that were available at that time were too scanty, no serious generalization could be made. Furthermore, the vast majority of then reported studies suggested that accumulation of an intermediate during protein unfolding was regarded as an exception to the rule, whereas a conformational transition described by a two-state model represented the “normal” response of a protein to changes in its environment. Although proteins were shown to respond differently to various kinds of denaturation, the supposition of the singleness of the denaturation process persisted [46].

For the first time, an intermediate state accumulating during the unfolding process was identified as early as in 1973 by Tanford’s group while looking at the chemical unfolding of bovine carbonic anhydrase B (BCAB) by GdmHCl [53]. It is notable that the intermediate state identified by far- and near-UV circular dichroism (CD) spectroscopy was described as having the secondary structure of the native state but as having lost the tertiary structure [53]. A year later, Kin-Ping Wong and Larry M. Hamlin used circular dichroism, difference spectrophotometry, enzymatic activity, and viscosity to study acid denaturation of these proteins and showed that the denatured acid BCAB was enzymatically inactive and did not have a unique 3D structure as judged by near-UV CD; it also did not exist in the random-coiled state as indicated by viscosity and far-UV CD [54]. Around the same time, Pititsyn’s group [27] initiated their work, which eventually led to further early insights into the folding intermediate. It was suggested that the formation of a native-like secondary structure preceded the proteins acquiring their tertiary structures. The results of the analysis of acid- and temperature-induced denaturation from this group were found to support this notion [10,11,13]. It was Ohgushi and Wada who in 1983 coined the term “molten globule” to describe such folding intermediates [12].

The most defining characteristics of a “classic” MG are outlined below [14,15,16,25,55,56,57,58,59,60,61,62,63,64,65]. A protein molecule in the MG state is characterized by the presence of a significant secondary structure (which is often classified as native-like secondary structure) with no or little tertiary structure (tight packing of side chains of amino acid residues is absent). Furthermore, 2D-NMR coupled with a hydrogen-deuterium exchange showed that the protein molecule in the MG state is characterized not only by the native-like secondary structure content, but also by the native-like folding pattern [66,67,68,69,70,71,72,73,74,75]. Small-angle X-ray scattering (SAXS) analysis revealed that the molten globular proteins possess globular structure typical of native globular proteins [76,77,78,79,80,81]. In agreement with the preservation of globular structure, the protein molecule in this state is characterized by a high degree of compactness, as its expansion typically leads to a general increase of 10–20% in radius of gyration or a hydrodynamic radius (over the native state), which corresponds to the volume increase of ~50% [57,82].

A considerable increase in the accessibility of a protein molecule to proteases was noted as a specific property of the MG state [83,84,85,86,87,88,89]. There was also an increase in the solvent exposure of the hydrophobic core, which was now less compact than the core of a native globular protein. This was reflected in the characteristic capability of the MG to specifically bind a hydrophobic fluorescent probe 1-anilino-naphthalene-8-sulfonate (ANS) or 1,1′-Bis(4-anilino-5-naphthalenesulfonic acid) (bis-ANS) [90,91,92]. MGs can show substantial levels of structure in some cases [55]. Lynne Regan reported that one part of a protein can retain the native structure, whereas another part forms an MG [93]. That is expected as proteins in general are characterized by noticeable structural heterogeneity, and conformational stability/flexibility can vary across the protein regions [94,95,96]. The abundant existence of intrinsically disordered proteins (IDPs) with various levels of disorder, and the presence of intrinsically disordered protein regions (IDPRs) in numerous proteins serve as extreme examples of this phenomenon [83,94,95,96].

While earlier data on the denaturation/unfolding and refolding of small proteins were compatible with the two-state model comprised of N → D and D → N transitions, the fact that many proteins were shown to form MGs during their unfolding indicated that the reality was more complex, and one should consider protein unfolding as the sequential process N ↔ MG ↔ U. This clearly raised a question on the physical and thermodynamic nature of the corresponding N ↔ MG and MG ↔ U transitions. The answer to this question was retrieved first from the results of the multiparametric experimental analysis of equilibrium GdmHCl-induced unfolding of BCAB and *S. aureus* β-lactamase at 4 °C, which clearly showed that the molten globule was separated from the more unfolded states by the “all-or-none” transition (this was evidenced by the bimodal distribution function of the molecular dimensions within the transition from the molten globule to the unfolded state) [97].

Later, similar bimodal distribution in the HPLC gel-filtration profiles was observed within the unfolding pathways of the NAD^+^-dependent DNA ligase from the thermophile *Thermus scotoductus* [22,23]. In line with these observations, an analysis of then available data on the equilibrium urea- and GdmHCl-induced N → U, N → MG, and MG → U transitions of globular proteins revealed that the cooperativity of all these unfolding processes increased linearly with the increase of the molecular weight of the protein up to 25–30 kDa. This indicated that the solvent-induced transitions from the native to the unfolded state, from the native to the molten globule state, and from the molten globule to the unfolded state were characterized by an “all-or-none” nature, thereby suggesting that the molten globule represented a third thermodynamic state of a protein molecule [98,99]. The validity of this model was later supported by Vijay S. Pande and Daniel S. Rokhsar, who in 1998 analyzed the equilibrium properties of proteins with Monte Carlo simulations and showed that, in addition to a rigid native state and a nontrivial unfolded state, a generic phase diagram contained a thermodynamically distinct MG state, further supporting the idea that MG represented a third phase state of proteins [100].

## 3. Potential Functionality of Folding Intermediates

Even before the acknowledgement of the prevalence and biological importance of intrinsically disordered proteins with their considerable structural heterogeneity, it was recognized that folding intermediates, including MGs, might have biological relevance. One of the first notes about this scenario was a hypothesis that the MG state may be involved in the translocation of proteins across membranes [101]. This idea was successfully supported by experiments, and there is now enough evidence that translocation of proteins and their insertion into membranes involve the MG state [102,103,104,105,106]. Model systems with α-lactalbumin showed the binding of MG to lipid bilayers [107]. In general, globular proteins can be transformed into the MG states on interaction with the membrane surface [108]. Such N → MG transitions in the vicinity of a membrane can be induced by the action of the so-called “membrane field”, which is a combination of the local decrease in the effective dielectric constant of water near the organic surface with the effect of negative charges located on the membrane surface [109,110,111,112]. Release and loading of the large, tightly packed hydrophobic ligands from and to the globular proteins might be facilitated by the partial unfolding of the carrier (N → MG transition) resulting from the concerted action of the moderate local decrease of pH and of the dielectric constant in proximity to the target membranes [113].

Furthermore, many proteins responsible for the transport of large hydrophobic ligands might have MG properties in their preloaded apo-forms [114,115,116]. It was also shown that many carbohydrate- and amino acid-binding periplasmic protein in *E. coli* form molten globule, which bind to their respective ligands [117]. Chaperonins interact with MGs and prevent their aggregation [118]. Earlier, Martin et al. discussed how a chaperonin-mediated folding had an MG as an intermediate [119]. It was also pointed out that compact, MG-like intermediates are localized within a central cavity of the chaperonin GroEL [120,121,122]. Facilitated folding of actins and tubulins occurs via a nucleotide-dependent interaction between the cytoplasmic chaperonin and the distinctive folding intermediates [123]. The presence of MG during nascent peptide folding has been inferred [124].

Importantly, although aforementioned functionalities have been attributed to the MG-like conformations, the major emphasis of all these and similar studies was still focused on the assumption that these functional MGs were folding intermediates kinetically trapped by the chaperonins just after the protein biosynthesis but before proteins become completely folded [25,101,125] or appear as a result of point mutations preventing polypeptides from complete folding [25,126] or originate from the denaturing effects of the membrane field [101,102,103,104,105,106,107,108,109,110,111,112] or ligand binding or release [114,115,116]. However, the presence of MGs in the cells become an established fact. [127]. All these observations provided strong support to the validity and importance of the concept of MG as a folding intermediate of globular proteins in vivo.

## 4. How Can One Find Molten Globules, and Where Can They Be Found?

MGs of globular proteins are generally obtained by their mild denaturation that can be induced by acid, alkali, low to medium concentrations of chemical denaturants such as urea and GdmHCl, chaotropic salts, moderately high temperature, and, for some proteins, even by low temperature [128,129,130,131,132,133,134,135,136,137,138,139,140,141,142,143,144,145,146,147]. Later studies revealed that in some proteins, an MG can also be induced by various organic solvents [148,149,150,151]. However, it was also shown that fluorinated alcohols can preferentially stabilize α-helices leading to the formation of non-native helical structures in some all-β-sheet proteins. For example, such highly helical states were induced by 2,2,2-trifluoroethanol (TFE) in several all β-sheet proteins, such as cardiotoxin analogue II (CTX II), from the Taiwan cobra (*Naja naja atra*) [152], procerain, a cysteine protease from Calotropis procera [153], β-lactoglobulin [154,155,156,157] and mellitin, [154,157] to name a few. All β-sheets to mostly α-helical structure in β-lactoglobulin and mellitin were also induced by hexafluoroisopropanol (HFIP), as well as by non-fluorinated alcohols, isopropanol, ethanol, and methanol [154,157]. Curiously, it was pointed out that an alcohol-induced α-helical state of β-lactoglobulin structurally resembles a transiently populated folding intermediate with high levels of non-native α-helical structure, which is formed within a few milliseconds during the refolding of this protein [158], suggesting that an intermediate with the non-native α-helical structure can accumulate during the refolding process of β-lactoglobulin, emphasizing that the hierarchical model cannot correctly describe folding of some β-structural proteins, including β-lactoglobulin [156,158].

The secondary and tertiary structures were evaluated generally by far- and near-UVCD, respectively. Secondary structure can also be evaluated with Fourier-transform infrared spectroscopy (FTIR) or optical rotatory dispersion (ORD), whereas viscosity measurements, gel-filtration chromatography, dynamic light scattering (DLS), SAXS, and electron microscopy are used to track expansion of the molecular volume [63,64]. The decrease in the compactness accompanied by the increased solvent accessibility of the hydrophobic core is normally estimated by looking at the binding of the fluorescent dye ANS to a protein molecule [90,91,92]. However, it was also pointed out that since ANS and bis-ANS have a strong affinity to the partially folded MG state, they can shift the equilibrium from favoring the native state (N) to favoring the MG state [91]. As a result, the apparent destabilization of the native state is observed, as was shown for the nucleotide-binding chaperonin DnaK [91]. On the other hand, binding of ADP or ATP to the native state of this protein resulted in a shift of the equilibrium from the MG toward the N state [91]. Furthermore, as early as 1995, Anthony L. Fink (1943–2008) cautioned that “It is important to note that the presence of ANS tends to increase the propensity of molten globules and compact denatured states to aggregate, and that aggregation increases the ANS fluorescence emission” [64].

Some other techniques like hydrogen-deuterium exchange, NMR, X-ray, isothermal titration calorimetry (ITC), differential scanning calorimetry (DSC), and computational methods have also been increasingly applied in later years [73]. In general, all the techniques/methods applicable to looking at protein structure and stability can give valuable information about partially folded intermediates like MGs [159]. For example, various fluorescence techniques, such as analysis of the intrinsic and extrinsic fluorescence (both steady-state and time-resolved), fluorescence anisotropy, Förster resonance energy transfer (FRET), dynamic and static fluorescence quenching, and proteolytic susceptibility are also used quite often [160].

In additional to classical examples of α-lactabumin, BCAB, and β-lactamase, both equilibrium and kinetic (transient) MGs have been described for a number of proteins and their mutants [161,162,163]. One interesting comparison is between the MGs formed by α-amylases from a thermophile and those formed from a mesophile [164]. This analysis revealed that the MG of the thermophile was more stable, which is not surprising. The polyols were less effective in refolding of the MG of the mesophilic enzyme [164].

Another interesting class of proteins are from halophiles. These generally require >0.5 M KCl to be functional. In several cases, these proteins just like those from thermophiles are fairly stable towards unfolding. The mechanism of halo-adaption was investigated by Gloss et al. [165] by looking at the kinetics of folding of urea denatured dihydrofolate reductases (DHFR) from *E. coli* and a halophile. In both cases, after a burst intermediate, formation of two intermediates was detected. The data was consistent with salt ions destabilizing the unfolded states in both cases. The authors concluded that halo-adaption involves affecting the solvent via a hydrophobic effect, the Hofmeister effect, preferential hydration, and crowding. This is in line with the X-ray crystallography and structural data that showed extensive solvation but little salt binding in the case of many halophilic proteins [165].

Yet another example of complexity in halo-adoption by halophile proteins is the role of protein hydration [166]. Given their higher surface charge density, it is widely believed that these are highly hydrated even in their native forms. This excessive hydration was expected to be responsible for the exceptional stability of corresponding proteins under saline conditions. The results obtained with an engineered protein with a high number of acidic residues on its surface suggested that not only was the surface hydration of a halophilic protein not much larger than that of a mesophilic counterpart, but even its hydration dynamics during unfolding was not very different [166].

Study of the proteasome from the extremely halophilic archaeon *Haloarcula marismortui* revealed that while other enzymes unfolded under sub-saline conditions, the proteasome was more resistant [167]. The biological significance of this is that it underlines how proteasome degrades the damaged proteins under sub-saline conditions as the stress situation for the organisms [167].

Uversky compared the stabilities of proteins from mesophiles with those from halophiles, thermophiles, and barophiles while advancing a hypothesis about the role of protein dielectricity in affecting the solvent properties in the context of protein-protein interactions [168]. The article mentions the earlier work with β-lactoglobulin, in which it was reported that the molten globule formation by the protein in alcohol-cosolvent mixtures was directly dependent on the decrease in the dielectric constant of the water as a result of mixing the simple alcohols [111]. Interestingly enough, in an independent observation, Gupta et al. around the same time observed that for a number of proteins, the enzyme stability in aqueous-organic cosolvent mixtures was dictated by the polarity index of the organic solvent [169]. Solvents with a polarity index of 5.8 and above were good cosolvents, which did not destabilize the protein even when up to 50% (*v*/*v*) is added to the aqueous buffer [169]. Both dielectric constants and polarity indexes are measures of solvent polarity.

Another interesting observation has been reported about MG formed by chymotrypsinogen [170]. A single cysteine reacts with glutathione at a very rapid rate. Such hyperactive cysteine residues are also present in serum albumin, lysozyme, and ribonuclease [170]. However, cysteine present in two proteins of a thermophile (in which glutathione is absent) did not display this hyper-reactivity. The authors infer that this unusually high reactivity of cysteine residues is relevant to the oxidative refolding of proteins in the organisms, which have oxidized glutathione-reduced glutathione system [170].

Furthermore, many IDPs exist as MGs under physiological conditions, and hence many important biological functions of such proteins, including cell signaling and other regulatory activities depend upon these molten globular states [56,69,171,172,173,174,175,176,177,178,179,180,181,182,183,184,185,186].

## 5. Baroenzymology, Cryoenzymology and Molten Globules

While the effects of the temperature on protein conformation are widely known, the influence of pressure on protein structure and function has also attracted considerable attention and is referred to as baroenzymology [187]. The effect of pressure on protein refolding has been especially intriguing and was discussed in a recent book [188]. Masahiro Watanbe et al. [189] used ultraviolet spectroscopy to compare the effect of pressure on native and molten globules of canine milk lysozyme with the corresponding behavior of the homologous protein bovine α-lactalbumin (BLA). Notably, the MG state of the lysozyme was found to have a more compact hydrophobic core; unlike the “swollen hydrophobic core of the MG state of BLA” [189]. This is an interesting result in the context of concepts of DMG and WMG, which now are commonly accepted kinds of molten globules.

High hydrostatic pressure at 600 MPa was shown to induce MG formation in another model protein, β-lactoglobulin, where this pressure-induced MG remained stable for at least three months [190]. High pressure induced a native dimer to a molten globule monomer transition in Arc repressor [191] and lactate dehydrogenase (LDH) [192], as well as promoting disassembly of the cowpea mosaic virus (CPMV) capsid into the molten globular monomers [193,194]. This is the case for many other proteins as well, such as trypsin [195], carboxypeptidase Y [196], butyrylcholinesterase (BuChE) [197], staphylococcal nuclease (SNase) [198], horse liver alcohol dehydrogenase (HLADH) [199], human Q26 and murine Q6 ataxin-3 [200], human serum albumin (HSA) [201], human acetylcholinesterase (hAChE) [202], and X-prolyl dipeptidyl aminopeptidase from *Streptococcus thermophilus* [203]. Taken together, all these data clearly indicate that high hydrostatic pressure has the unique property of stabilizing partially folded states or MG states of a protein [204].

Again, while heat denaturation is quite well acknowledged, cold denaturation of proteins is not so extensively mentioned in enzymology. Yet, this phenomenon has long been known [205,206,207,208]. For example, cold denatured states were described for myoglobin [206], a mutant of phage T4 lysozyme [209], α-lactalbumin [210,211], equine β-lactoglobulin [212], ubiquitin [213], and cytochrome c [214]. To prevent freezing of the aqueous buffered solution at subzero temperatures and to assist destabilization, organic solvents are also generally present in order to study cold denaturation. Kumar et al. discussed at length the cold denaturation of horse ferricytochrome c at extreme pH [215]. During acidic denaturation in the presence of anions, the partially folded state of the protein is referred to as an A state. Similarly, the partially folded state obtained under alkaline conditions in the presence of cations as counterions is referred to as B state. Although the A state and corresponding structural transitions have been studied in several cases, Kumar et al. have mentioned that the analysis of the B state has attracted much less attention [215].

## 6. Molten Globules and Intrinsic Disorder in Proteins

Coming back to the hypothesis on the potential role of protein dielectricity in affecting the solvent properties mentioned earlier [168], in the context of functional relevance of partially unfolded protein intermediates, it was proposed that a protein lowers the dielectric constant of the local medium around its interface with the aqueous solvent/water rich medium. This facilitates the behavior of proteins acting as “unfoldases”. Many proteins, in order to be functional, have to be unfolded (then referred to as conditionally disordered proteins) [216,217]. In many cases, this conditional unfolding is initiated by the interacting protein, which acts as an unfoldase by lowering the local dielectric around it; this leads to the binding between the two as a part of a biologically relevant process. Examples include unfolding of BCL-xL while interacting with the intrinsically disordered PUMA, which in turn folds upon binding as entropic compensation [168]. Unfoldases also include ATP-dependent proteases (such as in proteomes) and molecular chaperonins. Early examples in which this unfoldase behavior was observed were pore-forming domains of some toxins and carrier proteins of large nonpolar ligands. The aggregation including where it leads to amyloid formations (and is responsible for many diseases) may also be initiated by protein lowering the dielectric around it. Few other examples relevant to this are available [217,218,219,220]. Therefore, this hypothesis provides a common thread running through diverse phenomena [168]. Interestingly enough, later work has confirmed that functionally relevant unfolded structures of many bacterial toxins are molten globules [221,222,223].

One should keep in mind that intrinsic disorder in proteins represent a highly heterogeneous phenomenon, and functional IDPs can be disordered to different degrees. In fact, the existence of native (i.e., functional) coils, PMGs, and MGs was reported [56,181,183,184,185]. Furthermore, different parts of a protein molecule can be disordered to different degrees, and a functional protein can contain ordered, molten globular, pre-molten globular, and coil-like domains. What’s more, IDPs/IDPRs (and, as a matter of fact, any protein molecule in general) can be structurally represented as a spatio-temporal combinations of foldons (independent foldable units of a protein), inducible foldons (disordered regions that can fold at least in part due to the interaction with their binding partners), inducible morphing foldons (disordered regions that can fold differently at interaction with different binding partners), semi-foldons (always semi-folded regions), non-foldons (non-foldable protein regions), and unfoldons (regions that undergo an order-to-disorder transition to become functional) [94,95,224,225]. Another important note is that these functional disordered elements (i.e., foldons, inducible foldons, inducible morphing foldons, semi-foldons, and non-foldons) can structurally exist as coils, PMGs, or MGs.

There is another pointer to the complexity of the process. Bychkova et al. have discussed the differences between an MG and an IDP [127]. In the latter, there is a greater disruption of local structure; H-D exchange is higher. However, we do not have any data regarding a comparison between the two different forms of MGs (WMG and DMG) and IDPs.

There is also an interesting observation that MG-like IDPs can drive liquid-liquid phase separation (LLPS) that leads to the formation of protein condensates [226]. It is reported that in the case of the replication transcription of respiratory syncytial virus that take place within the “viral factories”, which are liquid-like structures within the cytosol of infected cells, the phosphoprotein tetramer (which is involved in the process and has a highly disordered N-terminal domain and a molten globular C-terminal domain) displays LLPS during a thermal transition, which is accompanied by the folding of the MG domain [226]. When the phosphoprotein is mixed with a nucleoprotein, which is also a part of the viral replication complex, again a phase separation is observed. Based on their observations, the authors of this study concluded that for LLPS to take place in vitro and in the cell, a weak, MG-like structure must be present, and such a structure defines physicochemical grounds for the LLPS behind the viral replication factory [226]. This is an interesting observation, as more often, proteins driving LLPS are expected to be either native coils (as shown for many IDPs [227,228,229,230,231,232,233]) or native PMGs (see, e.g., data for the AB region of human retinoid X receptor subtype γ (hRXRγ) [234]).

## 7. Engineered Molten Globules

Recently, one of us described a number of examples of engineered proteins which form molten globules [83]. These examples give a good idea of what kind of amino acid sequences favor formation of molten globules. Some of those examples are briefly recalled below.

Dihydrofolate reductase (DHFR, E.C.1.5.1.3) binds to the structural analogs of the substrate dihydrofolate. These inhibitors, such as methotrexate and trimethoprim, are well-known antifolate drugs. Mutants (Thr35Asp; Thr35Asp/Asn37Ser/Arg57His) of DHFR existed as MGs, which were catalytically active even though the mutations were made in the active site of the enzyme [235]. The binding of trimethoprim and NADPH to the MGs converted these mutants to a stable conformation close to the one obtained with the native enzyme [235]. Even more extensive mutations were carried out by circular permutation via linking the N- and C-termini of DHFR by a tripeptide and creating the new termini elsewhere [236,237]. One such circularly permutated variant existed as an MG, but addition of the ligands (substrates or antifolates) induced the MG → N transition [237].

An engineered Chorismate mutase from *Methanococcus jannascii* (MjCM) had the same *k_cat_* as the native enzyme [238,239,240,241,242], though the *K_m_* value with substrate chorismate was increased 3-fold [238]. The mutant possessed all the properties of an MG [239], and the induced-fit binding of the substrate took place on the same time scale as the native enzyme [240]. The native enzyme is a dimer whereas this mutant was a monomer. For this MG, the conformations in equilibrium were higher in number as compared to the native enzyme. The stopped flow kinetic studies showed that on and off rates of a mutant form were higher than those of the native enzyme [241]. All these data fit well to the model, where the less structured monomeric conformation of MjCM represents a catalytic MG.

The same group also designed a variant of MjCM with an amino acid composition consisting of just nine amino acids [243]. This variant with considerably reduced diversity in amino acid composition (the native enzyme sequence is made up of 19 different amino acids) was also an MG but was more structured (with higher helical content) than the mutant described above. This engineered protein is a good system to evaluate the value of diversity in amino acid composition in native enzymes [243].

Another interesting system studied under this approach has been glutathione transferase A1-I (GSTA1-1), a promiscuous enzyme critically important in detoxification [244,245]. Its remarkable catalytic promiscuity results from its molten globular structure. It exists as a rather broad conformational ensemble, wherein the conformations are freely interconvertible [246]. Curiously, GSTA4-4, another glutathione transferase of this α-class is highly specific [247,248,249]. This raised the question of how GSTA1-1 detoxifies structurally diverse substrates. An extensive study of many variants of both the GSTA1-1 and GSTA4-4 enzymes suggested that the active site structurally acts as a molten globule with rest has a well-defined structure [246].

The 5-Aminolevulinate synthase [ALAS] is the first enzyme of heme biosynthesis. A pyridoxal phosphate-dependent enzyme, it catalyses the condensation reaction between glycine and succinyl CoA. An engineered murine ALAS is described which forms a catalytically active MG but with considerably reduced *k_cat_* [250].

A transient folding intermediate during refolding of unfolded acylphosphatase from the archeon *Sulfolobus solfataricus* (Sso AcP) had an active site, which had little structure but did have enzyme activity [251]. Further refolding of this intermediate led to an ensemble of conformations with properties of an MG [252]. About 10% of molecules of this MG folded more slowly (than the rest) to the native states due to the requirement of cis-trans isomerisation of Leu49-Pro50 peptide bond. This is another example of incompletely folded yet catalytically active protein conformations [251,252].

Staphycoccal nuclease (SNase) was modified to form a deletion mutant, from which nine amino acid residues from both ends were removed. The resulting Δ131Δ mutant was mostly unfolded yet retained the activity of the native enzyme. The variant folded to the native state when high concentrations of the substrate was present [253,254]. NMR analysis revealed that the mutant (in the unfolded form) had retained most of the secondary structure elements of the wild-type protein, though these had far more flexibility [253,254,255,256].

A later study examined a double mutant F34W/W140F of this nuclease. This was found to exist as an MG, which could be folded to the native state [257]. The *K_m_* values of this MG for DNA, calcium ion, and pdTp were quite similar to that of the native enzyme. Its *V_max_* was also mostly unaltered. This indicated that the MG acquired the folded/native conformation in the presence of ligands/substrate [257].

Other interestingly engineered MGs were obtained by “de-evolution” of the dephospho-CoA kinase (DPCK) from *Aquifex aeolicus* by substituting its aromatic amino acids (including histidines) with Leu residues or non-aromatic amino acids based on the best preservation of thermodynamic stability [258]. The four variants created had about 10 % of the protein sequence altered. Two variants (DPCK-LH and DPCK-M), unlike the wild-type enzyme, displayed ATPase activity in the absence of dCoA. Moreover, the DPCK-LH also showed phosphotransferase activity in the presence of dCoA. The variants had secondary structures similar to those of the wild enzyme, but contained less tertiary structure, indicating that these were MGs [258].

Another example of the catalytic engineered MGs is given by “artificial, rationally designed catalytic polypeptides” termed oxaldies [259]. These 14 amino acid residue-long polypeptides were capable of the spontaneous formation of the amphiphilic α-helix and self-oligomerization to a four-helix bundle and higher order aggregates, and they had a reactive amine anchored to them [259]. Notably, these MGs were able to catalyze oxaloacetate decarboylation rather efficiently [259].

These examples show that MGs probably are extreme cases of flexible proteins with adequate potential for catalysis left at least partially undisturbed. The terms inactivation and denaturation refer to the unfolding process viewed from the different windows of biological activity and structure. Catalytic MGs underline this notion.

## 8. Macromolecular Crowding and Molten Globules

It is realized that the intracellular environments are vastly different from the dilute solutions of enzymes in aqueous buffers, which are mostly used in enzymology. The intracellular volume is crowded with diverse macromolecules [260,261], where the concentrations of proteins, polysaccharides, and nucleic acids (in free or conjugated forms) are estimated to be in a range of 80–400 mg/mL [262,263,264,265], and where diverse macromolecules and small molecules together occupy about 40% of the cytosolic volume [266]. A large number of studies have tried to simulate these environments by adding macromolecules (such as “inert” polymers, e.g., Poly(ethylene glycol) PEG, Dextran, Ficoll, and Poly(sodium 4-styrene sulfonate) (PSS) of different molecular mass and some “inert” proteins, such as bovine pancreatic trypsin inhibitor (BPTI), ribonuclease A, lysozyme, β-lactoglobulin, hemoglobin, bovine serum albumin (BSA)), and high concentrations of low molecular weight substances [40,261,267]. The following examples discuss the implications of crowding on the formation of MGs.

A molten globule of apo-myoglobin was obtained by using high salt concentrations from the pH-denatured protein [268]. Being utilized as a crowding agent, dextran stabilized this MG against unfolding by both heat and cold. CD was used to look at the transitions and the results agreed with the excluded volume theory [268]. Similarly, the acid-unfolded form of cytochrome c at pH 2.0 was shown to undergo folding transition to the MG state after addition of high concentrations of dextran to protein solution [269]. On the other hand, a recent study showed that even under the physiological condition of pH 7.0 and 25 °C, cytochrome c adopted the MG in the presence of high PEG-400 concentrations [270]. This crowding-induced N → MG transition was attributed to the soft interactions between PEG and protein, indicating that macromolecular crowding effects are more complex than the excluded-volume [270].

More recently, it has been shown that Ficoll-70 interacts with the heme group of myoglobin, which converts the protein into an MG at physiological pH [271]. An FTIR study of the C-terminal domain of histone H1 in the presence of crowders PEG and Ficoll showed that this IDP becomes folded and gains noticeable levels of regular secondary structure [272]. However, fluorescence studies revealed that this is actually an MG. Perhaps, similarly, formation of this MG under the crowded conditions of the cell “may increase the rate of the transition towards the DNA bound state and facilitate H1 diffusion inside cell nuclei” [272]. Alkali pH-unfolded ferricytochrome c and lysozyme at pH 12.9 (±0.1) were shown to adopt MG conformations in the presence of various crowding agents (Dextran-40, Dextran-70, and Ficoll-70) [273].

The MGs of α-lactalbumin (LA) are easily obtained by subjecting its calcium ions’ depleted form (apo-form) to different denaturing conditions [274]. An MG of the apo-form has been shown to play a role in apoptosis of tumor cells [275]. It was reported that the presence of PEG-2000 as a crowding agent also could lead to the formation of an MG by human apo-LA [276]. Curiously, the other two polymeric crowding agents tried, Ficoll-70 and dextran-70, did not have this unfolding effect [276]. An ITC analysis revealed that while these two crowders did not interact with the human apo-LA, PEG had a weak non-specific interaction with the protein [276]. Presumably, this weak interaction was sufficient to overcome the stabilizing effect of the crowder due to the excluded volume [276].

Atomistic MD simulations of three proteins with different structural organization (an intrinsically disordered 47-residue activator for thyroid hormone and retinoid receptor [ACTR], a molten globular 51-residue nuclear coactivator-binding domain of CREB [NCBD)], and an ordered 191-residue interferon regulatory transcription factor [IRF-3]) were performed to assess the effect of small-sized synthetic (PEG500) and protein crowders (at concentrations of 175–300 g/L) on the structure, dynamics, and interactions of these query proteins [277]. The results showed that the degree of disorder in a protein plays a critical role in its response to crowding. The excluded volume effects pushed the conformation towards compact structure; quinary (weak transient) interactions favored extended conformation [277]. Interestingly, while crowding slowed down protein flexibility and restricted the conformational landscape, thereby resulting in a bias toward the “bioactive conformations”, it simultaneously diminished the biologically relevant interactions. Another important finding was that PEG 500 as the synthetic crowder had different consequences than protein-induced crowding [277].

## 9. Interactions of Nanomaterials with Molten Globules

The foundations of our understanding of protein adsorption onto non-porous materials were laid by Wilem Norde [278]. Perceptively, he stated that proteins can be either hard or soft in the context of their structural changes upon adsorption. The latter kind undergoes significant structural changes upon adsorption. With the advent of nanoscience, the interactions of proteins with diverse nanomaterials has been extensively studied [279]. Proteins bound to the nanoparticles are referred to as their corona. The importance of the corona in drug design/delivery has been discussed in a number of places [280,281]. In 2013, Saptarshi et al. have mentioned that the human carbonic anhydrase when bound to silica nanoparticles forms an MG [280], whereas detaching of nanoparticles resulted in the formation of three catalytically active intermediates with “native-like structures” [280].

In a large number of cases, conformational changes/unfolding of proteins upon binding to the nanoparticles have been documented [280]. Carbonic anhydrase is one of the most extensively studied proteins in the context of molten globules. Billsten et al. looked at the adsorption of one of its mutant and two truncated forms, in which 4 and 16 amino acids had been deleted from the N-terminal on silica nanoparticles [282]. While the whole-length mutant did not show any structural change, the truncated forms were present as MGs on the nanoparticles. These MGs were very similar to the MG formed when the whole mutant enzyme was unfolded by GdmHCl [282].

Furthermore, in some instances, interactions with nanoparticles were shown to induce fibrillation [283]. It has been observed that HbAO, the major component of human blood hemoglobin, formed an MG upon interaction with copper nanoparticles, which led to the protein aggregation [284]. This did not happen with HbA2, a protein isoform associated with β-thalassemia. The authors suggested that this behavior could form a basis for screening for thalassemia [284]. An excellent review discussing how the nanoparticle properties along with the protein nature influences the outcome of their interaction in terms of conformational changes leading to MGs and/or fibrils is available [285].

## 10. Dry and Wet Molten Globules

An excellent paper published by Denisov et al. in 1999 focused on the hydration of non-native states of proteins. Early SAXS, DLS, and heat capacity data suggested that MGs have “substantial internal hydration” [286]. Disruption of tertiary structure during the MG formation is expected to lead to the increased hydration of protein interior, as water molecules are believed to become competing H-bond partners. ^1^H nuclear Overhauser effect (NOE) spectroscopy and ^17^O magnetic relaxation dispersion (MRD) data for a few structurally unrelated proteins (α-lactalbumin, lysozyme, ribonuclease A, apomyoglobin, and carbonic anhydrase) provided a finer picture. It was found that the MG states and native structures of these proteins have comparable levels of internal hydration, suggesting that the MG forms of these proteins are more structured and less solvent exposed than commonly believed [286].

Dry molten globules (DMGs) are molten globules characterized by the low hydration and efficient shielding of the hydrophobic core from the solvent [34]. Thus, these precede WMGs during unfolding and are structurally closer to the native state in general, though these have all other characteristics commonly ascribed to MGs. DMGs do have expanded volumes, and their formation is accompanied by the conformational unlocking of the side chains (and related gain of the conformational entropy), though liquid-like van der Waals interactions are still present. The U → DMG step is a major free energy barrier in the entire U → N transition due to the large enthalpic contribution, though DMG is stabilized by the compensation via an increase in the conformational entropy. While WMGs can be detected by fluorescence and near-UV CD, DMGs (with almost no water invading to change microenvironments of Trp and other side chains), these techniques can miss the DMG formation. Their detection can be done by NMR and FRET techniques [34]. However, analysis of the refolding kinetics could track the DMG → N transition in the case of RNAse A by the ratio test, which measures kinetics by both tracking secondary structure by far-UV CD and tertiary structure by near-UV CD [34]. In cases of both dihydrofolate reductase (DHFR) and monellin (a sweet protein that has two noncovalently associated polypeptide chains: a 44-residue-long A chain, and a B chain with 50 residues), it was shown that ANS dye was bound to WMG but not to DMG [34].

Although early reports on the existence of DMGs started appearing in 1995 [38,287], it was not until late 2000 that these intermediate states were accepted as real intermediates, which are distinct from the WMG (which is considered a classic MG now) on folding/unfolding pathways [35,36,288]. The extended simulation of the unfolding of lysozyme B urea at 37 °C caught the formation of DMG, though a similar study at a higher temperature had failed to do so, indicating that the higher temperature destabilized the DMG [288]. Urea (unlike water) interacted with the peptide backbones [288].

Looking at the unfolding of barstar and its mutants with several techniques has revealed some finer details about formations of DMG and WMG [28]. In this study, near-UV CD indicated that urea-induced unfolding started with the loss of tertiary contacts to form an intermediate N*. FRET showed that this early intermediate expanded to form I. Fluorescence spectral measurements showed that both of these intermediates, N* and I were DMGs. Dynamic quenching of the single buried Trp in core suggested the later formation of WMG. The slowest step in the unfolding process of barstar was the unfolding of WMG, and it had a solvated transition state [28].

The villin headpiece subdomain HP35 has been studied by triplet-triplet energy transfer, and locked (native state) and unlocked states (DMG) were identified [32]. The DMG was characterized by a solvent-free core but showed increased flexibility and “local unfolding” behavior. High pressure triplet-triplet energy transfer measurements revealed that while increasing pressure (which favors the N → DMG transition) was not accompanied by any expansion, the reverse transition showed a volume change. This indicated the existence of two DMG forms, where one is as compact as a native structure and the other has an expanded volume [32].

In the case of multidomain proteins, it is possible that not all domains are in DMG states [30,289]. A recent review focusses on DMGs and their roles in diseases [29].

## 11. Pre-Molten Globule States

The term pre-molten globule was coined in 1991 by Mei-Fen Jeng and S. Walter Englander to refer to a folding intermediate, which was observed during the unfolding of cytochrome c and which was less structured than even MGs [290]. Here, cytochrome c molecules at low pH and sodium chloride concentration of <0.05 M were found to expand beyond the MG state as seen by the viscosity and fluorescence. While the helical contents are nearly all retained, “tertiary structural hydrogen bonds are largely broken (hydrogen exchange rates), some normally buried parts of protein are exposed to water (fluorescence) and many of the native side chain contacts must be lost” [290].

In the same year, Chaffotte et al. were looking at a C-terminal peptide F2 of the β-2 subunit of *E. coli* tryptophan synthase using a number of biophysical techniques [291]. They concluded that “neither the secondary nor the tertiary structure of isolated F2 resembled those of native F2. In this respect, isolated F2 is not a molten globule” [291]. A few years later, the same laboratory found that transient intermediates formed within 2–4 msec during refolding of several proteins gave different estimates of the secondary structure contents with different techniques [292]. Again, looking more closely into the case of the F2 fragment, they found that the isolated F2 folded into a “condensed, but not compact” conformation [292]. This conformation was in rapid equilibrium with the conformations with the native and non-native secondary structures, and it was described as a pre-molten globule (PMG) [292].

In 1994, Uversky and Ptitsyn reported that β-lactamase at a low temperature and in the presence of GuHCl formed two partially folded intermediates, the classic MG and a new equilibrium state of protein molecules, which they originally called a “partly folded” state [26]. However, in 1996, the same group described a four-state GdmHCl-induced unfolding of BCAB at a low temperature [24] and concluded that in both proteins, unfolding is described as a four-state process, N ↔ MG ↔ PMG ↔ U. Furthermore, in these studies, PMG was found to have an expanded volume of “no more than two-fold” and had solvent-exposed clusters of nonpolar amino acids. It was found that significant levels of secondary structure content is retained at this stage of unfolding. Similar results were earlier reported with beta-lactamase as well [24,26]. Therefore, cryoenzymology could be used to detect the formation of PMG.

When the salt sodium sulphate was gradually added to the unfolded Barstar (present under highly alkaline conditions), the first PMG of the protein was observed [37]. At a higher concentration, PMG got converted to MG, which was a dry MG as no water was present in the core. This dry MG had about 65% secondary structure and 40% tertiary contacts (as compared to the native form). It was also shown that this MG was a productive intermediate on the folding pathway [37]. The solvation dynamics around the active site of glutaminyl-tRNA synthetase was found to increase during the transition from MG → PMG [293]. Both partially folded intermediates had similar levels of secondary structure, though PMG was more flexible. Furthermore, DLS revealed that in both cases, protein aggregates were present [293].

A series of further papers have confirmed this four-state picture comprising a pre-molten globule (PMG) and an MG as intermediates during folding or unfolding of several proteins. Georlette et al. showed that the equilibrium GdmCl-induced unfolding of the NAD+-dependent DNA ligase from the thermophile *Thermus scotoductus* follows the four-state N → MG → PMG → U model, where, similar to BCAB and β-lactamase, the MG → PMG transition was characterized by the presence of a bimodal distribution of the molecular dimensions in HPLC gel-filtration profiles, indicating that this process represents an “all-or-none” transition [22,23].

Khan et al. carried out experiments on the unfolding and refolding of a mutant Leu94Gly of cytochrome c, which was earlier recognized as an MG [294,295]. Denaturation with LiCl led them to identify a PMG of the mutant, which was less stable than the mutant by about 5.4 KCal/mole and more stable than the unfolded protein by merely about 1.1 KCal/mole [294,295].

It has been shown that myoglobin in the presence of PEG of intermediate sizes formed PMG [296]. This was probably the first report of the formation of a PMG under physiological conditions but in the presence of a crowding agent. Thus, it is likely that both MGs and PMGs are formed under overcrowded conditions [296].

In another example, cytochrome c under acidic conditions was found to form an MG and a PMG when glucose and dextran-70 were used as crowders [297]. The authors believe that the polymeric crowder stabilized the protein [297].

It is likely that with more powerful techniques, we will find more than three (PMG, WMG, and DMG) intermediates en route to folding of the ordered proteins. In some way, the work of Garcia-Fandino et al. [298] is a pointer to the validity of this hypothesis. To gain detailed information on the mechanism of the temperature-induced unfolding of apoflavodoxin, these researchers carried out atomistic multi-microsecond-scale molecular dynamics (MD) simulations. It is remarkable to see many intermediates as early as 10 microseconds into the unfolding process (Figure 1).

## 12. Conclusions and Future Perspectives

Quite often, our insights about the protein folding process have come by looking at protein unfolding! It started with protein denaturation; the data about that segues well with studies on protein stability. In general, protein stability under one stress condition is often accompanied by stability under other denaturing conditions. The protein unfolding starts with denaturation of the polypeptide chain followed by numerous physicochemical changes [299].

With numerous biological phenomena, we have learnt that a two-step model is often the result of our inability to “see” multiple steps in the process. As our tools become more powerful and we develop better computational tools, we become able to see multiple intermediates. Curiously, this is also true in enzyme kinetics. Steady state kinetics was available first. As fast kinetic methods became available, we had pre-steady state kinetics and more detailed pictures of “on” and “off” processes [300]. So, it is not surprising that two-state models of protein unfolding have increasingly been replaced by four-state models, where during the unfolding process, protein undergoes sequential transitions from the native state to the molten globule, then to the pre-molten globule before eventually reaching the unfolded state. Thermal denaturation does not produce totally unfolded conformation; one needs some other kind of stress like a chemical denaturant to complete unfolding. The same is applicable to most proteins in extremely acidic and basic conditions. This means that we have more than one kind of denatured state. The fast accumulating data on the wet and dry MGs further indicate that the journey is not over yet, and we still are on the road. Recently, trapping the co-populated protein conformers during acid-induced unfolding of cytochrome c, myoglobin, and lysozyme indicated the existence of finer details of the unfolding process [301]. Of special interest is the “conformational shuttling”, in which the population of an unfolded form of cytochrome c at a low pH first increases and then decreases with time [301].

Since deep mutational scanning represents a fast and convenient way of acquiring knowledge on the residue-specific contribution to protein interactions involving IDPs, it is likely that such approaches will be able to further shed light on ways by which molten globules play a biological role beyond being only folding intermediates [302].

The classical way has viewed the unfolding/folding processes purely from the blinkered view of structure. With increasing understanding about the role of intrinsic disorder, this is slowly opening up another dimension [41]. However, we still do not have a clear and comprehensive answer to a question on all of the roles these various MGs and PMGs may play under in vivo conditions. After all, crowded intracellular conditions seem to affect protein conformation. Another unexplored aspect is related to specificity of these different unfolded (or differently partially folded) forms. Are they promiscuous (in more or different ways in comparison with the ordered globular forms) [303]? Do they play a role in protein evolution? Are they involved in moonlighting [304]? Do they play a role in immune responses [305]? What are the various ways we can use them in applied biocatalysis or drug delivery designs [306,307,308,309,310,311,312]? We already know that the partially unfolded states of some proteins are precursors to aggregation, and many diseases originate from aggregation via intrinsic disorder [225,313,314,315,316,317,318,319,320,321]. Let us not forget that unfolding increases disorder. Also, the roles of flexible conformations in protein assemblies and protein-protein interactions is likely to be more important than our explorations have revealed so far [322].

This review, by describing a thread running through different phenomena/approaches, such as cryoenzymology, baroenzymology, macromolecular crowding, intrinsic disorder, and interactions of partially unfolded proteins with nanomaterials hopefully stimulate further research into the various facets of molten globules.

## Figures and Tables

**Figure 1 ijms-24-02424-f001:**
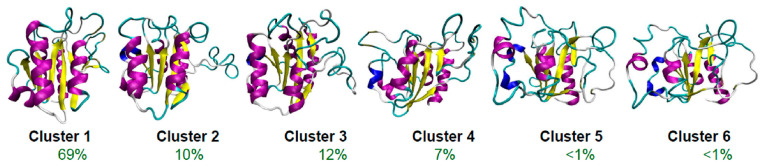
Early intermediates in the thermal unfolding of apoflavodoxin. The percentage of population of these clusters is displayed. Modified from García-Fandiño R, Bernadó P, Ayuso-Tejedor S, Sancho J, Orozco M (2012) Defining the Nature of Thermal Intermediate in 3 State Folding Proteins: Apoflavodoxin, a Study Case. PLoS Comput Biol 8(8): e1002647 [298], which is an open-access article distributed under the terms of the Creative Commons Attribution License, which permits unrestricted use, distribution, and reproduction in any medium, provided the original author and source are credited.

## Data Availability

Not applicable.

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
