# Peer review of "Pre-Molten, Wet, and Dry Molten Globules en Route to the Functional State of Proteins"

_ijms, 2023, doi:10.3390/ijms24032424_

Round 1

Reviewer 1 Report

Understanding the intricacies of protein folding transition states and formation of unfolding intermediates is a complex area of research. Uversky et al. have produced extremely valuable insights into the mechanisms of protein folding. This particular paper focuses on pre-molten, wet, and dry molten globules long understood to be involved directly in protein structure and function development. In this review, both experienced Authors extensively presented an extremely comprehensive review of the publications on the protein unfolding intermediates.

There is some doubt about the title of the review, as only in the 10th paragraph are the properties of the WMG and DMG discussed. The review contains as many as 311 citations, although the PubMed database lists only 10 referring to WMG and 39 to DMG. Undoubtedly, these conformational states of proteins are worthy of research and popularization. Also, the historical background is interesting and above all, is a description of the molten globule as the third thermodynamic state of biologically relevant protein molecules.

It seems that the Authors should devote a little more space to the relationship between MG, PMG, and WMG, DMG, respectively,  and unfolding routes U-WMG-DMG-N and U-MG -PMG-N. Furthermore, they could pay more attention to IDP/IDRs and their relation to the unfolded states of the protein molecule. It would be interesting for readers to know the Authors' opinions on the structural causes of droplet formation in the phase separation process (LLPS) by proteins containing unstructured fragments. Is there any link to MG, WMG or DMG states.

This review is well-written and adds value to the field of molten globule properties.

Minor points:

4oC  - line 179 ; superscript ?

{ref}  - line 635 ; lack of reference

Author Response

Understanding the intricacies of protein folding transition states and formation of unfolding intermediates is a complex area of research. Uversky et al. have produced extremely valuable insights into the mechanisms of protein folding. This particular paper focuses on pre-molten, wet, and dry molten globules long understood to be involved directly in protein structure and function development. In this review, both experienced Authors extensively presented an extremely comprehensive review of the publications on the protein unfolding intermediates.

There is some doubt about the title of the review, as only in the 10th paragraph are the properties of the WMG and DMG discussed.

Response: Thank you for pointing this out. In fact, we have a brief discussion of the different forms of molten globules already in the second paragraph of the Introduction section. We also added a scheme showing correlations between different protein forms.

The review contains as many as 311 citations, although the PubMed database lists only 10 referring to WMG and 39 to DMG. Undoubtedly, these conformational states of proteins are worthy of research and popularization. Also, the historical background is interesting and above all, is a description of the molten globule as the third thermodynamic state of biologically relevant protein molecules.

It seems that the Authors should devote a little more space to the relationship between MG, PMG, and WMG, DMG, respectively,  and unfolding routes U-WMG-DMG-N and U-MG -PMG-N.

Response: Thank you for pointing this out. A scheme has been inserted to summarize the relationships.

Furthermore, they could pay more attention to IDP/IDRs and their relation to the unfolded states of the protein molecule. It would be interesting for readers to know the Authors' opinions on the structural causes of droplet formation in the phase separation process (LLPS) by proteins containing unstructured fragments. Is there any link to MG, WMG or DMG states.

Response: Thank you for pointing this out. The corresponding discussion was added to section “6. Molten globules and intrinsic disorder in proteins”. Discussion of emerging role of MGs in LLPS has been added as well.

This review is well-written and adds value to the field of molten globule properties.

Minor points:

4oC  - line 179 ; superscript ?

{ref}  - line 635 ; lack of reference

Response: Thank you, we have carefully gone through the manuscript. The corrections you indicated have been incorporated.

Reviewer 2 Report

The manuscript "Pre-molten, wet, and dry molten globules en route to the functional state of proteins" is a very extensive and well-referenced review into the topic of protein folding intermediates. It offers an interesting historical perspective as well as comprehensive analysis of recent data and discoveries. A minor issue is the complete lack of figures to illustrate some of the concepts and break up a very dense block of text. These might take the form of cartoons illustrating different types of intermediates, their interconversion, and perhaps an associated table summarizing their characteristics. Another cartoon might illustrate various functions of MGs. There are a number of minor grammatical issues that would benefit from careful English revision. I have listed a few typos for correction below:

1. Line 56, I think "new" should be "now"

2. Line 57, "or" should be deleted

3. Line 139, "proteins" should not be plural

4. Line 166, "substantially" should be "substantial"

5. Line 191 "he" should be "the"

6. Line 274 "X-ray" is listed as a technique. Please be more specific (e.g. SAXS, crystallography, etc.)

7. Line 396 should read "give a good idea..."

8. Line 402 delete "formed"

9. Lines 431-432 consider revising. Maybe something like "...the active site structurally acts as a MG while the rest has a well-defined structure."

10. Line 456 replace "Yet another" with "Other"

11. Line 516 seems to be missing the word "interact"

12. Line 562 appears to have the beginning of an unnecessary sentence. Consider revision

13. Line 567 should use either "several" or "a few", not both.

14. Line 635 replace [ref] with the appropriate reference

15. Line 666. I assume this should read "...was less stable than the native..."

Author Response

The manuscript "Pre-molten, wet, and dry molten globules en route to the functional state of proteins" is a very extensive and well-referenced review into the topic of protein folding intermediates. It offers an interesting historical perspective as well as comprehensive analysis of recent data and discoveries. A minor issue is the complete lack of figures to illustrate some of the concepts and break up a very dense block of text.

Response: Thank you for pointing this out. The text now at many places been further broken into shorter paragraphs. A fig and a scheme have been added.

These might take the form of cartoons illustrating different types of intermediates, their interconversion, and perhaps an associated table summarizing their characteristics. Another cartoon might illustrate various functions of MGs.

Response: Thank you for pointing this out. We have added a scheme showing their interconversion. Individual characteristics of these forms of MGs vary somewhat in individual cases. Hence, it is felt that at this stage, creation of a table with general characteristics may be a little premature or an oversimplification. For that reason, we have avoided giving such a table and limited ourselves to describing their respective features in individual cases.

There are a number of minor grammatical issues that would benefit from careful English revision. I have listed a few typos for correction below:

  1. Line 56, I think "new" should be "now"
  2. Line 57, "or" should be deleted
  3. Line 139, "proteins" should not be plural
  4. Line 166, "substantially" should be "substantial"
  5. Line 191 "he" should be "the"
  6. Line 274 "X-ray" is listed as a technique. Please be more specific (e.g. SAXS, crystallography, etc.)
  7. Line 396 should read "give a good idea..."
  8. Line 402 delete "formed"
  9. Lines 431-432 consider revising. Maybe something like "...the active site structurally acts as a MG while the rest has a well-defined structure."
  10. Line 456 replace "Yet another" with "Other"
  11. Line 516 seems to be missing the word "interact"
  12. Line 562 appears to have the beginning of an unnecessary sentence. Consider revision
  13. Line 567 should use either "several" or "a few", not both.
  14. Line 635 replace [ref] with the appropriate reference
  15. Line 666. I assume this should read "...was less stable than the native..."

Response: Thank you for pointing this out. Apart from the first one (where new is what we meant), all corrections have been incorporated. We have also gone through the manuscript more carefully and made couple of corrections elsewhere.